# Associations of treated and untreated human papillomavirus infection with preterm delivery and neonatal mortality: A Swedish population-based study

Johanna Wiik[1,2,3]*, Staffan Nilsson[4,5], Cecilia Kärrberg[1,3,6], Björn Strander[1,6], Bo Jacobsson[1,3,7], Verena Sengpiel[1,3]

1 Department of Obstetrics and Gynaecology, Institute of Clinical Sciences, Sahlgrenska Academy, University of Gothenburg, Gothenburg, Sweden, 2 Department of Gynaecology and Obstetrics, Østfold Hospital Trust, Kalnes, Norway, 3 Department of Obstetrics and Gynaecology, Sahlgrenska University Hospital, Gothenburg, Sweden, 4 Department of Mathematical Sciences, Chalmers University of Technology, Gothenburg, Sweden, 5 Department of Laboratory Medicine, Institute of Biomedicine, Sahlgrenska Academy, University of Gothenburg, Gothenburg, Sweden, 6 Regional Cancer Centre West, Region Västra Götaland, Gothenburg, Sweden, 7 Department of Genetics and Bioinformatics, Division of Health Data and Digitalisation, Norwegian Institute of Public Health, Oslo, Norway

* johanna.wiik@gu.se

**Data Availability Statement:** Data cannot be shared publicly because the dataset is a collection of data from several Swedish national registers and

## Abstract

### Background

Treatment of cervical intraepithelial neoplasia (CIN) is associated with an increased risk of preterm delivery (PTD) although the exact pathomechanism is not yet understood. Women with untreated CIN also seem to have an increased risk of PTD. It is unclear whether this is attributable to human papillomavirus (HPV) infection or other factors. We aimed to investigate whether HPV infection shortly before or during pregnancy, as well as previous treatment for CIN, is associated with an increased risk of PTD and other adverse obstetric and neonatal outcomes.

### Methods and findings

This was a retrospective population-based register study of women with singleton deliveries registered in the Swedish Medical Birth Register 1999–2016 ($n = 1,044,023$). The study population had a mean age of 30.2 years (SD 5.2) and a mean body mass index of 25.4 kg/m$^2$ (SD 3.0), and 44% of the women were nulliparous before delivery. Study groups were defined based on cervical HPV tests, cytology, and histology, as registered in the Swedish National Cervical Screening Registry. Women with a history of exclusively normal cytology ($n = 338,109$) were compared to women with positive HPV tests ($n = 2,550$) or abnormal cytology ($n = 11,727$) within 6 months prior to conception or during the pregnancy, women treated for CIN3 before delivery ($n = 23,185$), and women with CIN2+ diagnosed after delivery ($n = 33,760$). Study groups were compared concerning obstetric and neonatal outcomes by logistic regression, and comparisons were adjusted for socioeconomic and health-related confounders. HPV infection was associated with PTD (adjusted odds ratio [aOR] 1.19, 95%

we are not allowed to share it. The mandatory Swedish Medical Birth Register and the Swedish Cancer Registry are national datasets and therefore considered to be public property. Access to the data is given only to researchers with permission from a Swedish regional ethical review board (see https://www.etikprovningsmyndigheten.se) and after approval of the research plan by the data managers. Data access requests may be sent to the Swedish National Board of Health and Welfare (https://www.socialstyrelsen.se). After permission from a Swedish regional ethical review board, data researchers can also apply to access data from Statistics Sweden (https://www.scb.se) and the Swedish National Cervical Screening Registry (http://www.nkcx.se/).

**Funding:** JW has received research grants from Østfold Hospital Trust, Norway (nr 16/04196) https://sykehuset-ostfold.no/helsefaglig/forskning, and Hjalmar Svenssons forskningsfond, Sweden (nr HJSV2020079) https://www.stiftelsemedel.se/stiftelsen-handlanden-hjalmar-svenssons-forskningsfond/. VS has received research grants from Wilhelm och Martina Lundgren Vetenskapsfond (2016-1005) https://wmlundgren.se/ and Fru Mary von Sydows, född Wijk, donationsfond (nr 1016) https://www.maryvonsydowstiftelsen.se/. The funders had no role in study design, data collection and analysis, decision to publish, or preparation of the manuscript.

**Competing interests:** I have read the journal´s policy and the authors of this manuscript have the following competing interests: CK has research grants for clinical trial Assar Gabrielsson's Foundation for Cancer-Related Clinical Research, Hjalmar Svensson's Research Foundation. She is presently head of one Swedish site of a multicenter international academic clinical HPV-vaccine trial where the sponsor, Imperial College, London, receive some support from MSD. She is a member of the Screening and colposcopy national guidelines committee, and she is the secretary of The Swedish Society for Colposcopy and Cervical Cancer Prevention/C-ARG. She has received reimbursement from Gedeon Richter for lectures on meeting arranged by the company. BS has research grants for clinical trial by the Governmental agreement with Swedish counties, ALF, and by the Region of Västra Götaland. He is presently Swedish head of an multicenter international academic clinical HPV-vaccine trial where the sponsor, Imperial College, London, receive some support from MSD. The past five years he has served as chairman of the Swedish Cervical screening coordinating committee, as

CI 1.01–1.42, $p = 0.042$), preterm prelabor rupture of membranes (pPROM) (aOR 1.52, 95% CI 1.18–1.96, $p < 0.001$), prelabor rupture of membranes (PROM) (aOR 1.24, 95% CI 1.08–1.42, $p = 0.002$), and neonatal mortality (aOR 2.69, 95% CI 1.25–5.78, $p = 0.011$). Treatment for CIN was associated with PTD (aOR 1.85, 95% CI 1.76–1.95, $p < 0.001$), spontaneous PTD (aOR 2.06, 95% CI 1.95–2.17, $p < 0.001$), pPROM (aOR 2.36, 95% CI 2.19–2.54, $p < 0.001$), PROM (aOR 1.11, 95% CI 1.05–1.17, $p < 0.001$), intrauterine fetal death (aOR 1.35, 95% CI 1.05–1.72, $p = 0.019$), chorioamnionitis (aOR 2.75, 95% CI 2.33–3.23, $p < 0.001$), intrapartum fever (aOR 1.24, 95% CI 1.07–1.44, $p = 0.003$), neonatal sepsis (aOR 1.55, 95% CI 1.37–1.75, $p < 0.001$), and neonatal mortality (aOR 1.79, 95% CI 1.30–2.45, $p < 0.001$). Women with CIN2+ diagnosed within 3 years after delivery had increased PTD risk (aOR 1.18, 95% CI 1.10–1.27, $p < 0.001$). Limitations of the study include the retrospective design and the fact that because HPV test results only became available in 2007, abnormal cytology was used as a proxy for HPV infection.

## Conclusions

In this study, we found that HPV infection shortly before or during pregnancy was associated with PTD, pPROM, PROM, and neonatal mortality. Previous treatment for CIN was associated with even greater risks for PTD and pPROM and was also associated with PROM, neonatal mortality, and maternal and neonatal infectious complications.

## Author summary

### Why was this study done?

- Treatment of cervical intraepithelial neoplasia (CIN) is associated with an increased risk of preterm delivery (PTD), preterm prelabor rupture of the membranes (pPROM), and neonatal mortality. However, the exact biological mechanism is still unknown.

- Women with untreated CIN also seem to have an increased risk of PTD. It is unknown whether HPV infection can cause PTD or other adverse obstetric outcomes.

- Although ascending bacterial infection is an established leading cause of PTD, there is limited knowledge concerning whether PTD after excisional treatment for CIN is associated with infectious complications.

### What did the researchers do and find?

- This retrospective population-based register study (1999–2016) found that HPV infection shortly before or during pregnancy is associated with an increased risk of PTD, pPROM, prelabor rupture of the membranes, and neonatal mortality.

- Treatment for CIN was also associated with these adverse outcomes and with an even higher risk for PTD and pPROM.

- This study presents new evidence that previous treatment for CIN is associated with an increased risk of maternal and neonatal infectious complications.

chairman of the Screening and colposcopy national guidelines committee, and as head of the Cervical screening process register. He has been a member of the National Board of health and welfare evaluation committee for the Cervical screening program. He has received no financial support or reimbursement from commercial companies for any activity. BJ has research grants from the Swedish Research Council, the Research Council of Norway, the March of Dimes, and the Burroughs Wellcome Fund. During the last five years, he has performed clinical diagnostic trials for Ariosa, Natera, Vanadis, and Hologic with reimbursement costs per recruited patient. He has conducted clinical trials on probiotics in pregnancy in collaboration with BioGaia and FukoPharma. He has also been involved in the IMPACT study where Roche and Thermo Fisher paid for PLGF-analyzes. He has also arranged scientific meetings with commercial partners (ESPBC 2016) and a Nordic educational meeting on NIPT and preeclampsia screening (2017). No lectures, travel, or personal reimbursements from the companies.

**Abbreviations:** aOR, adjusted odds ratio; CIN, cervical intraepithelial neoplasia; HPV, human papillomavirus; LLETZ, large loop excision of the transformation zone; MBR, Swedish Medical Birth Register; NKCx, Swedish National Cervical Screening Registry; pPROM, preterm prelabor rupture of membranes; PROM, prelabor rupture of membranes; PTD, preterm delivery; SGA, small for gestational age.

## What do these findings mean?

- Although we cannot claim causality due to the register-based study design, our results suggest that pregnancies after treatment for CIN should be regarded as high-risk pregnancies and counseled accordingly.

- These results support the idea that strategies to mitigate HPV infection, such as vaccination programs, may be beneficial for maternal and neonatal pregnancy outcomes.

## Introduction

Human papillomavirus (HPV) infection is the most common genital infection in women of reproductive age. Persistent genital infection with high-risk HPV is causally associated with cervical cancer and its precursor, cervical intraepithelial neoplasia (CIN) [1]. The widespread introduction of cervical cancer screening [2,3] and subsequent treatment of CIN has lowered the incidence of cervical cancer significantly [4]. Unfortunately, surgical excision to treat CIN has been associated with an increased risk of preterm delivery (PTD), preterm prelabor rupture of membranes (pPROM), conditions requiring neonatal intensive care, and neonatal mortality in subsequent pregnancies [5]. PTD, birth before 37 weeks of gestation, is an enormous global obstetric and societal problem, as it is the main cause of mortality in children under the age of 5 years, as well as of short- and long-term morbidity [6,7].

The mechanism behind excisional treatment increasing the risk of PTD remains unclear. The hypotheses include increased risk of ascending bacterial infection, HPV-induced immunomodulation, and acquired "mechanical weakness" secondary to loss of cervical tissue [8]. Although bacterial infection is an established major cause of PTD, especially early PTD and PTD starting with pPROM [9], there is limited knowledge concerning whether PTD after excisional treatment is associated with infectious complications [10].

The magnitude of the increase in risk for PTD attributed to treatment differs between studies [8,10–15] and seems to depend on the type of treatment. The risk increase is higher after excision than after ablation and is highest after cold knife excision [10]. Risk of PTD also seems to increase with the depth of excision [10]. Furthermore, the risk increase depends on whether healthy women or women with untreated CIN/HPV infection serve as the comparison group [10]. While it has been suggested during recent years that untreated CIN also confers a risk of PTD [8,16,17], a recent Cochrane review and meta-analysis concluded that further research is needed to understand the causal link and possible pathomechanism [10]. It is as yet unclear whether the pathomechanism is the HPV infection itself or whether the association between CIN and PTD is due to a vulnerability to both conditions, either by genetic predisposition or as a product of confounding [10,16]. It is thus unknown whether HPV infection can cause PTD or other adverse obstetric outcomes. Studies linking positive HPV tests and/or abnormal cervical cytology with obstetric outcomes have shown conflicting results [18–21]. A recent meta-analysis and systematic review suggested an association between HPV infection and PTD, and possibly pPROM [22], although all included studies were small. Other obstetric or neonatal outcomes in women with untreated CIN/HPV infection have not been studied in detail [10,17,23,24].

In summary, the risk of PTD increases after treatment for CIN, but the underlying mechanism is still not known. Women with untreated CIN also seem to have an increased risk of

PTD, but it has not been established whether this risk is attributable to HPV infection during pregnancy. Neither the associations between untreated HPV infection in pregnancy and PTD and other obstetric and neonatal outcomes, nor the association between previous treatment for CIN and infectious pregnancy complications, have as yet been studied in a comprehensive population-based study. In this large Swedish national population-based study, we aimed to investigate whether HPV infection shortly before or during pregnancy is associated with an increased risk of PTD and other adverse obstetric and neonatal outcomes. We also aimed to study the association between previous treatment for CIN and PTD and other adverse obstetric and neonatal outcomes, with special focus on infectious complications at birth.

## Materials and methods

### Data sources

This study is a retrospective population-based study using data from different Swedish health and population registers. Data linkage between the registers was performed with the unique personal identification number held by each resident of Sweden. The Swedish Medical Birth Register (MBR), established in 1973 and managed by the Swedish National Board of Health and Welfare, is the object of compulsory registration and thus comprises all live births and stillbirths in Sweden at 22 completed weeks of gestation and up [25]. The information available in the register is extracted from antenatal and delivery unit medical records and includes data on maternal health, reproductive history, pregnancy complications and neonatal outcomes. The Swedish National Cervical Screening Registry (NKCx) contains data from 1978 onwards, with full national coverage of all cervical cytology results since 1997 and full national coverage of histology results since 1998 [26]. Cervical HPV tests have been recorded in the database since 2007. The Swedish Cancer Register, a mandatory registry established in 1958 and managed by the Swedish National Board of Health and Welfare, registers all diagnosed cancer cases in Sweden, as well as all cases of CIN3 [27]. Information on education level, country of birth, and income was obtained from the Swedish Register on Participation in Education [28], the Total Population Register [29], and the Income and Tax Assessment Register [30], managed by Statistics Sweden.

This study is reported as per the Strengthening the Reporting of Observational Studies in Epidemiology (STROBE) guideline (S1 STROBE Checklist). The study was prospectively planned in 2015, the dataset was received in 2019, and analyses were conducted until November 2020. We did not publish the analysis plan, but the overall exposure, outcomes, confounders, and analyses were planned in 2015 by the research team based on hypotheses drawn from previous studies. After obtaining and reviewing the content of the dataset, but before starting the analyses, the final definition of study groups, outcomes, and study period was decided based on the available data and quality of the dataset. During the analysis process, we decided to also conduct stratified analyses.

### Study population

All women with singleton births between 1 January 1999 and 31 December 2016 were identified in the MBR. Women with a history of chronic inflammatory disease, human immunodeficiency virus infection, or organ transplantation were excluded (see S1 Table for the International Classification of Diseases [ICD] codes leading to exclusion), leaving a study population comprising 1,787,842 deliveries in 1,044,023 women with a mean age at delivery of 30.2 years (SD 5.2) and a mean body mass index (BMI) of 25.4 kg/m$^2$ (SD 3.0); 44% of the women were nulliparous before delivery. Of these, 400,583 women had at least 1 delivery that fulfilled the criteria for inclusion in a study group (see below; Fig 1).

## Exposures

A delivery registered in the MBR was eligible for inclusion in a study group if the woman fulfilled the criteria for exposure, i.e., results of a cervical HPV test registered in the NKCx in 2007–2016, cervical cytology or histology registered in the NKCx in 1978–2016, and/or a histological diagnosis registered in the Swedish Cancer Register. For exact classification of study groups, see S2 Table. For classification of cytology and histology results, see S3 Table.

A reference group was defined, consisting of women who had a history of exclusively normal cervical cytology results until the end of the study period and at least 1 sample taken in the 3 years preceding the included delivery ($n = 338,109$).

The exposure groups were defined as follows:

1. HPV infection group: women with presumed HPV infection during pregnancy, based on abnormal cervical cytology or positive HPV test recorded within 6 months prior to conception or during pregnancy. Two partly overlapping subgroups were defined:

    1a. HPV infection (cytology): abnormal cytology ($n = 11,727$).

    1b. HPV infection (HPV test): positive cervical HPV test ($n = 2,550$).

2. Subsequent CIN2+ group: at presumed risk of persistent HPV infection based on a histological diagnosis of CIN2 or CIN3, adenocarcinoma in situ (AIS), or cervical cancer any time after delivery ($n = 33,760$).

3. Treated group: histologically diagnosed CIN3 before conception. Since there is no national Swedish register covering treatment for CIN, CIN3 was used as a proxy for treatment as these women are always treated in Sweden, a policy that has been consistent over the study period ($n = 23,185$).

In order to avoid women previously treated for CIN being included in the HPV infection groups or the subsequent CIN2+ group, women with histologically diagnosed CIN1 more than once or with CIN2+ before the pertinent delivery were excluded from these groups.

Three different treatment periods, based on the year of first diagnosis of CIN3, were defined, corresponding to differences in predominant mode of treatment: 1978–1985 (cold knife conization), 1986–1995 (tissue-saving methods such as cryotherapy, laser vaporization, laser conization, and diathermy), and 1996–2016 (large loop excision of the transformation zone [LLETZ], also referred to as loop electrosurgical excision procedure [LEEP]) [31].

In the treated group, the interval from treatment to delivery was determined, as was the interval from delivery to CIN2+ diagnosis in the subsequent CIN2+ group.

## Outcomes

The primary outcome was PTD at 22–36 weeks of gestation (154–258 days). Gestational age was retrieved from the MBR using the best estimate, i.e., ultrasound determination when available and last menstrual period or estimation of gestational age at delivery ward in the remaining cases. Subanalyses for early PTD (22–33 weeks, or 154–237 days, of gestation) and for very early PTD (22–27 weeks, or 154–195 days, of gestation) were performed.

Secondary outcomes were pPROM (determined according to ICD-10 codes in the MBR) and spontaneous PTD (defined as pPROM or preterm labor, excluding deliveries that started with induction or cesarean section, according to ICD-10 codes and MBR parameters).

Further studied outcomes comprised prelabor rupture of membranes (PROM) in term pregnancies (≥37 weeks of gestation), chorioamnionitis, intrapartum fever, neonatal sepsis, Apgar score < 7 at 5 minutes, intrauterine fetal death, neonatal mortality (1–28 days), and

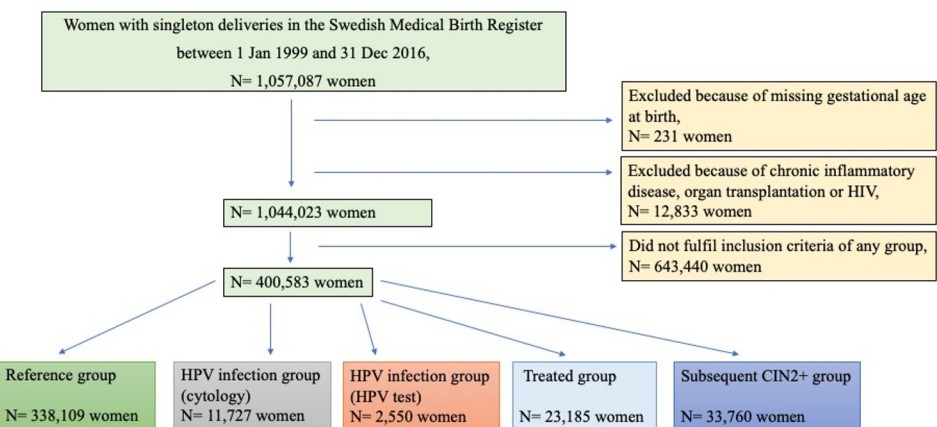

**Fig 1. Flowchart of the study population.** CIN, cervical intraepithelial neoplasia; HIV, human immunodeficiency virus; HPV, human papillomavirus.

small for gestational age (SGA), defined as birthweight less than −2 SD according to the Swedish reference curves [32] (S4 Table).

## Background variables

The following background variables were retrieved from the MBR and from Statistics Sweden registers: age at delivery, BMI, smoking before and during pregnancy, marital status, education level at delivery, employment at delivery, disposable household income in the 3 years preceding delivery, country of birth, parity, infant's sex, and assisted reproduction, as well as chronic renal disease, diabetes, epilepsy, and chronic hypertension as reported in antenatal care records.

## Statistical analyses

Descriptive data are presented as number and percentage for categorical variables and as mean (SD) and median (interquartile range) for continuous variables. Analyses were performed using R (version 4.0.0; R Foundation for Statistical Computing, Vienna, Austria; https://www.r-project.org/) and SPSS software (version 26.0; IBM; https://www.ibm.com/analytics/spss-statistics-software). A significance level of 0.05 was applied throughout.

The study groups were compared concerning obstetric and neonatal outcomes by unadjusted and adjusted logistic regression analysis. If a woman had several deliveries that fulfilled the criteria for inclusion in a group, only 1 delivery was included. A random delivery in the reference group was compared to the first eligible delivery after treatment in the treated group and the last eligible delivery in the HPV infection groups (cytology and HPV test) and the last delivery before diagnosis of CIN2+ in the subsequent CIN2+ group. The first eligible delivery after treatment in the treated group was compared to the last delivery in a woman included in the other exposure groups. This was done to better match for age and parity, since women in the treated group naturally were older than women in the other exposure groups. Women who had both a delivery after treatment and a previous delivery included in any of the other exposure groups were excluded from the treated group when the treated group was compared to the other exposure groups.

HPV infection can exist for a long time before diagnosis of CIN2+. Therefore, the subsequent CIN2+ group was compared to the reference group after stratification for interval (0–3 years or >3 years) to diagnosis of CIN2+, based on the median interval from delivery to CIN2+ diagnosis of 3.2 years. The 2 groups were also compared with each other.

Women with a delivery included in the subsequent CIN2+ group followed by a delivery included in the treated group were included in paired analyses with conditional logistic regression analysis.

When comparing the HPV infection (HPV test) group with other groups, only deliveries after 2006 were included, since no HPV tests were registered before that year. In order to test whether cytology was a valid substitute for a positive HPV test, the 2 HPV infection groups were compared to each other. In these analyses, women with both a positive HPV test and abnormal cytology were excluded from the HPV infection (cytology) group.

The multivariable analyses were adjusted for potential confounding factors, including background variables with an uneven distribution between the groups, i.e., year of delivery (1999–2001, 2002–2004, 2005–2007, 2008–2010, 2011–2013, 2014–2016), maternal age (<21, 21–25, 26–30, 31–35, 36–40, >40), parity (0, 1, 2, 3, >3), BMI (underweight [<18.5 kg/m$^2$], normal weight [18.5–24.9 kg/m$^2$], overweight [25.0–29.9 kg/m$^2$], obese ≥30.0 kg/m$^2$], missing), marital status (cohabiting, single, other, missing), country of birth (Sweden, Europe, Asia, America/Oceania, Africa, missing), infant's sex (boy, girl, missing), smoking (never, before pregnancy, in early pregnancy, in the third trimester, missing), highest disposable household income in the 3 years preceding delivery (population divided into tertiles for every year), education level (primary, secondary, post-secondary < 3 years, post-secondary ≥ 3 years, missing), and assisted reproduction (yes, no). The paired analyses were adjusted for BMI, marital status, infant's sex, smoking, and assisted reproduction, with the group definitions defined above.

In order to study the implication of infectious complications, the treated group was compared with the reference group regarding chorioamnionitis and neonatal sepsis, with adjusted logistic regression after stratification for PTD. The treated group was also compared with the reference group regarding chorioamnionitis and neonatal sepsis in PTD cases, after stratification for pPROM and also adjusted for gestational age.

Neonatal mortality in the treated and HPV infection groups was compared with that in the reference group, with adjusted logistic regression analysis also including PTD in the adjustments. Subgroup analysis comparing neonatal mortality in the treated and reference groups was also performed, after stratification for infectious complications (chorioamnionitis and/or neonatal sepsis) and also adjusted for gestational age.

In all stratified analyses, differences in odds ratios between strata were analyzed by an interaction term between the stratum and the group in logistic regression.

### Ethics statement

The study was approved by the Regional Ethical Review Board at the University of Gothenburg (registration number 614–15). In this retrospective register-based study, no consent was obtained from the participants since the data were analyzed anonymously.

## Results

### Characteristics of the study population

Background factors for the study population are presented in Table 1. The study groups differed, for example regarding age, parity, and smoking. In the reference group, 10% (n = 32,457 women) were older than 35 years, and similarly 10% in the subsequent CIN2+ group (n = 3,492) and 12% in the HPV infection (cytology) group (n = 1,366) were older than 35 years, while age over 35 years was more frequent in the HPV infection (HPV test) group (n = 429, 17%) and the treated group (n = 4,755, 21%). Nulliparity was most frequent in the treated group (n = 13,425, 58%) and HPV infection groups (cytology, n = 6,479, 55%; HPV test, n =

**Table 1. Demographics and clinical characteristics in the study groups.**

| Characteristics | Reference group (n = 338,109) | HPV infection groups | | Treated group (n = 23,185) | Subsequent CIN2+ group (n = 33,760) |
|---|---|---|---|---|---|
| | | Cytology (n = 11,727) | HPV test (n = 2,550) | | |
| **Age at delivery (years)** | | | | | |
| <21 | 858 (0.3) | 161 (1.4) | 12 (0.5) | 10 (0.0) | 1,205 (3.6) |
| 21–25 | 54,778 (16.2) | 2,673 (22.8) | 445 (17.5) | 1,276 (5.5) | 7,402 (21.9) |
| 26–30 | 148,021 (43.8) | 4,447 (37.9) | 875 (34.3) | 7,865 (33.9) | 12,099 (35.8) |
| 31–35 | 101,995 (30.2) | 3,080 (26.3) | 789 (30.9) | 9,279 (40.0) | 9,562 (28.3) |
| 36–40 | 29,344 (8.7) | 1,177 (10.0) | 368 (14.4) | 4,095 (17.7) | 3,144 (9.3) |
| >40 | 3,113 (0.9) | 189 (1.6) | 61 (2.4) | 660 (2.8) | 348 (1.0) |
| **BMI class (kg/m$^2$)** | | | | | |
| Underweight (<18.5) | 6,886 (2.0) | 326 (2.8) | 79 (3.1) | 376 (1.6) | 882 (2.6) |
| Normal weight (18.5–24.9) | 193,930 (57.4) | 6,871 (58.6) | 1,587 (62.2) | 13,678 (59.0) | 19,233 (57.0) |
| Overweight (25–29.9) | 76,964 (22.8) | 2,461 (21.0) | 499 (19.6) | 5,153 (22.2) | 7,348 (21.8) |
| Obese (≥30) | 34,493 (10.2) | 1,160 (9.9) | 253 (9.9) | 1,868 (8.1) | 2,935 (8.7) |
| Missing | 25,836 (7.6) | 909 (7.8) | 132 (5.2) | 2,110 (9.1) | 3,362(10.0) |
| **Smoking** | | | | | |
| Never | 286,236 (84.7) | 8,582 (73.2) | 2,057 (80.7) | 16,735 (72.2) | 22,151 (65.6) |
| Before pregnancy | 25,166 (7.4) | 1,327 (11.3) | 260 (10.2) | 2,727 (11.8) | 4,415 (13.1) |
| In early pregnancy | 6,254 (1.8) | 530 (4.5) | 56 (2.2) | 1,079 (4.7) | 2,102 (6.2) |
| In the third trimester | 9,111 (2.7) | 941 (8.0) | 121 (4.7) | 1,732 (7.5) | 3,793 (11.2) |
| Missing | 11,342 (3.4) | 347 (3.0) | 56 (2.2) | 912 (3.9) | 1,299 (3.8) |
| **Infant's sex** | | | | | |
| Boy | 174,095 (51.5) | 6,108 (52.1) | 1,322 (51.8) | 11,853 (51.1) | 17,297 (51.2) |
| Girl | 164,000 (48.5) | 5,618 (47.9) | 1,228 (48.2) | 11,331 (48.9) | 16,462 (48.8) |
| Missing | 14 | 1 | 0 | 1 | 1 |
| **Assisted reproduction[1]** | 9,341 (2.8) | 153 (1.3) | 46 (1.8) | 934 (4.0) | 434 (1.3) |
| **Employment** | | | | | |
| Full-time | 181,520 (53.7) | 5,728 (48.8) | 1,458 (57.2) | 14,315 (61.7) | 14,719 (43.6) |
| Part-time | 79,475 (23.5) | 2,469 (21.1) | 438 (17.2) | 4,278 (18.5) | 8,390 (24.9) |
| None | 47,200 (14.0) | 2,388 (20.4) | 449 (17.6) | 2,359 (10.2) | 6,605 (19.6) |
| Missing | 29,914 (8.8) | 1,142 (9.7) | 205 (8.0) | 2,233 (9.6) | 4,046 (12.0) |
| **Parity** | | | | | |
| 0 | 163,533 (48.4) | 6,479 (55.2) | 1,481 (58.1) | 13,425 (57.9) | 12,730 (37.7) |
| 1 | 129,737 (38.4) | 3,407 (29.1) | 724 (28.4) | 5,968 (25.7) | 14,547 (43.1) |
| 2 | 36,004 (10.6) | 1,308 (11.2) | 257 (10.1) | 2,788 (12.0) | 4,826 (14.3) |
| 3 | 6,780 (2.0) | 354 (3.0) | 54 (2.1) | 751 (3.2) | 1,187 (3.5) |
| >3 | 2,055 (0.6) | 179 (1.5) | 34 (1.3) | 253 (1.1) | 470 (1.4) |
| **Marital status** | | | | | |
| Cohabiting | 312,277 (92.4) | 10,286 (87.7) | 2,252 (88.3) | 20,777 (89.6) | 29,551 (87.5) |
| Single | 3,625 (1.1) | 346 (3.0) | 63 (2.5) | 443 (1.9) | 1,007 (3.0) |
| Other | 7,369 (2.2) | 626 (5.3) | 126 (4.9) | 792 (3.4) | 1,605 (4.8) |
| Missing | 14,838 (4.4) | 469 (4.0) | 109 (4.3) | 1,173 (5.1) | 1,597 (4.7) |
| **Concurrent disease[2]** | | | | | |
| Renal disease | 1,326 (0.4) | 63 (0.5) | 17 (0.7) | 110 (0.5) | 184 (0.5) |
| Diabetes | 1,864 (0.6) | 69 (0.6) | 14 (0.5) | 104 (0.4) | 144 (0.4) |
| Epilepsy | 1,519 (0.4) | 66 (0.6) | 15 (0.6) | 121 (0.5) | 176 (0.5) |
| Chronic hypertension | 1,218 (0.4) | 39 (0.3) | 12 (0.5) | 99 (0.4) | 108 (0.3) |
| **Education level[3]** | | | | | |
| Primary school, up to 9 years | 20,947 (6.2) | 1,439 (12.3) | 234 (9.2) | 1,798 (7.8) | 4,820 (14.3) |
| Secondary | 136,656 (40.4) | 5,329 (45.54) | 967 (37.9) | 10,092 (43.5) | 17,386 (51.5) |

*(Continued)*

**Table 1.** (Continued)

| Characteristics | Reference group (n = 338,109) | HPV infection groups | | Treated group (n = 23,185) | Subsequent CIN2+ group (n = 33,760) |
| --- | --- | --- | --- | --- | --- |
| | | Cytology (n = 11,727) | HPV test (n = 2,550) | | |
| Post-secondary < 3 years | 46,648 (13.8) | 1,525 (13.0) | 358 (14.0) | 3,438 (14.8) | 4,007 (11.9) |
| Post-secondary ≥ 3 years | 132,012 (39.0) | 3,294 (28.1) | 971 (38.1) | 7,806 (33.7) | 7,148 (21.2) |
| Missing | 1,846 (0.5) | 140 (1.2) | 20 (0.8) | 51 (0.2) | 399 (1.2) |
| **Country of birth** | | | | | |
| Sweden | 290,269 (85.9) | 9,597 (81.8) | 2,053 (80.5) | 21,395 (92.3) | 29,724 (88.0) |
| Europe | 20,091 (5.9) | 872 (7.4) | 181 (7.1) | 1,017 (4.4) | 2,233 (6.6) |
| Asia | 19,105 (5.7) | 784 (6.7) | 188 (7.4) | 495 (2.1) | 1,213 (3.6) |
| America/Oceania | 3,253 (1.0) | 167 (1.4) | 47 (1.8) | 207 (0.9) | 381 (1.1) |
| Africa | 5,297 (1.6) | 297 (2.5) | 80 (3.1) | 70 (0.3) | 203 (0.6) |
| Missing | 94 (0) | 10 (0.1) | 1 (0.0) | 1 (0) | 6 (0.0) |
| **Year of delivery** | | | | | |
| 1999–2001 | 37,540 (11.1) | 1,346 (11.5) | 0 | 3,097 (13.4) | 6,389 (18.9) |
| 2002–2004 | 39,384 (11.6) | 1,250 (10.7) | 0 | 3,268 (14.1) | 6,372 (18.9) |
| 2005–2007 | 43,310 (12.8) | 1,198 (10.2) | 5 (0.2) | 3,368 (14.5) | 6,595 (19.5) |
| 2008–2010 | 53,847 (15.9) | 1,544 (13.2) | 76 (3.0) | 3,872 (16.7) | 6,566 (19.4) |
| 2011–2013 | 66,932 (19.8) | 2,533 (21.6) | 542 (21.3) | 4,259 (18.4) | 5,494 (16.3) |
| 2014–2016 | 97,096 (28.7) | 3,856 (32.9) | 1,927 (75.6) | 5,321 (23.0) | 2,344 (6.9) |
| **Highest disposable household income in 3 years preceding delivery** | | | | | |
| Lowest tertile | 44,834 (13.3) | 2,674 (22.8) | 610 (23.9) | 2,555 (11.0) | 6,320 (18.7) |
| Middle tertile | 100,738 (29.8) | 3,882 (33.1) | 758 (29.7) | 6,217 (26.8) | 11,007 (32.6) |
| Highest tertile | 192,535 (56.9) | 5,171 (44.1) | 1,181 (46.3) | 14,413 (62.2) | 16,426 (48.7) |
| **Gestational age estimation method** | | | | | |
| Ultrasound | 309,355 (91.5) | 10,828 (92.3) | 2,429 (95.3) | 20,836 (89.9) | 30,148 (89.3 |
| Last menstrual period | 15,200 (4.5) | 474 (4.0) | 45 (1.8) | 1,230 (5.3) | 2,024 (6.0) |
| Other[4] | 13,554 (4.0) | 425 (3.6) | 76 (3.0) | 1,119 (4.8) | 1,588 (4.7) |
| **Cytology diagnosis** | | | | | |
| Low grade | | 9,916 (84.6) | | | |
| High grade | | 1,811 (15.4) | | | |
| **Time period of CIN3 diagnosis** | | | | | |
| 1978–1985 | | | | 76 (0.3) | |
| 1986–1995 | | | | 2,467 (10.6) | |
| 1996–2016 | | | | 20,642 (89.0) | |
| **Interval from CIN3 diagnosis to delivery (years)** | | | | | |
| Mean (SD) | | | | 4.4 (3.4) | |
| Median (IQR) | | | | 3.4 (1.9–6.1) | |
| **Interval from delivery to diagnosis of CIN2/CIN3/AIS/cancer (years)** | | | | | |
| Mean (SD) | | | | | 4.3 (3.7) |
| Median (IQR) | | | | | 3.2 (1.4–6.2) |

Data are presented as number (percentage) unless otherwise specified. Percentages are based on those with available data. Percentages of missing are based on the total numbers.

[1] Treatment to achieve pregnancy.

[2] As reported in antenatal care records. Missing values were interpreted as the woman not having any concurrent disease, in accordance with how data are registered in antenatal care records.

[3] Highest education level at time of delivery.

[4] Ultrasound, last menstrual period, and/or estimation of gestational age at delivery ward.

AIS, adenocarcinoma in situ; BMI, body mass index; CIN, cervical intraepithelial neoplasia; IQR, interquartile range; SD, standard deviation.

1,481, 58%) and less frequent in the reference group (*n* = 163,533, 48%) and subsequent CIN2 + group (*n* = 12,730, 38%). Smoking during pregnancy was less common in the reference group (*n* = 15,365, 4.5%) than the exposure groups (HPV infection [cytology], *n* = 1,471, 12.5%; HPV infection [HPV test], *n* = 177, 6.9%; treated, *n* = 2,811, 12.1%; subsequent CIN2+, *n* = 5,895, 17.5%).

The comparisons presented below for different obstetric outcomes are adjusted logistic regression analyses; unadjusted analyses are presented in S5–S7 Tables.

## Preterm delivery

In the total dataset of 1,787,842 deliveries in 1,044,023 women, 87,727 deliveries (4.9%) were PTD and 62,951 deliveries were spontaneous PTD (3.5%).

Compared to the reference group (4.6%), the treated group had the highest risk of PTD (9.1%, adjusted odds ratio [aOR] 1.85, 95% CI 1.76–1.95, *p* < 0.001), but an increased risk was also found in the HPV infection (cytology) group (5.9%, aOR 1.21, 95% CI 1.12–1.31, *p* < 0.001) and the HPV infection (HPV test) group (5.6%, aOR 1.19, 95% CI 1.01–1.42, *p* = 0.042) (Table 2; Fig 2A). These results were similar for spontaneous PTD; the treated group had aOR 2.06 (95% CI 1.95–2.17, *p* < 0.001) and the HPV infection (cytology) group had aOR 1.18 (95% 1.07–1.29, *p* = 0.001) (Table 2; Fig 2B). The treated group and the HPV infection (cytology) group also had increased risk of very early and early PTD (Table 2).

Women in the subsequent CIN2+ group also had a slightly increased risk of PTD (5.1%, aOR 1.12, 95% CI 1.06–1.18, *p* < 0.001) and spontaneous PTD (Table 2). However, when it came to the analyses stratified for interval from delivery to diagnosis, the increase was only significant for women diagnosed with CIN2+ within the first 3 years after delivery (Table 3). In comparison with the other exposure groups (rather than with the reference group), the treated group exhibited an increased risk of both PTD and spontaneous PTD (Table 4; Fig 2).

## Preterm prelabor rupture of membranes

In the study population, 27,687 deliveries started with pPROM (1.5%). Compared to the reference group (1.5%), the risk of pPROM was increased in the treated group (4.0%, aOR 2.36, 95% CI 2.19–2.54, *p* < 0.001), the HPV infection (cytology) group (2.0%, aOR 1.22, 95% CI 1.07–1.40, *p* = 0.004), and the HPV infection (HPV test) group (2.5%, aOR 1.52, 95% CI 1.18–1.96, *p* = 0.001) (Table 2; Fig 3). Women in the subsequent CIN2+ group did not have an increased risk of pPROM compared to the reference group (Table 2), although an increased risk was found if the women were diagnosed with CIN2+ within the first 3 years after delivery (Table 3). Compared with the other exposure groups, the treated group had an increased risk of pPROM (Table 4; Fig 3).

## PROM in term pregnancies

Of the 1,700,115 term pregnancies, a total of 111,519 deliveries started with PROM (6.6%). Women in the treated group had an increased risk of PROM at term compared to the reference group (8.4% versus 6.8%, aOR 1.11, 95% CI 1.05–1.17, *p* < 0.001) (Table 2) and compared to the subsequent CIN2+ group (aOR 1.20, 95% CI 1.11–1.31, *p* < 0.001) (Table 4). Moreover, the HPV infection (HPV test) group had an increased risk of PROM compared to the reference group (aOR 1.24, 95% CI 1.08–1.42, *p* = 0.002). The associations for the HPV infection (cytology) group were oriented in the same direction but were not statistically significant (Table 2).

**Table 2. Obstetric and neonatal outcomes in exposure groups compared to the reference group—adjusted multivariable logistic regression analyses.**

| Outcome | Reference group (n = 338,109) | HPV infection groups | | | | | | Treated group (n = 23,185) | | | Subsequent CIN2+ group (n = 33,760) | | |
|---|---|---|---|---|---|---|---|---|---|---|---|---|---|
| | | Cytology (n = 11,727) | | | HPV test (n = 2,550) | | | | | | | | |
| | n (%) | n (%) | aOR (95% CI) | p-Value | n (%) | aOR[1] (95% CI) | p-Value | n (%) | aOR (95% CI) | p-Value | n (%) | aOR (95% CI) | p-Value |
| PTD, <37 weeks | 15,661 (4.6) | 692 (5.9) | 1.21 (1.12–1.31) | **<0.001** | 143 (5.6) | 1.19 (1.01–1.42) | **0.042** | 2,106 (9.1) | 1.85 (1.76–1.95) | **<0.001** | 1,736 (5.1) | 1.12 (1.06–1.18) | **<0.001** |
| Early PTD, <34 weeks | 4,221 (1.2) | 221 (1.9) | 1.39 (1.21–1.60) | **<0.001** | 34 (1.3) | 0.99 (0.70–1.40) | 0.96 | 661 (2.9) | 2.02 (1.86–2.21) | **<0.001** | 488 (1.4) | 1.18 (1.06–1.30) | **0.002** |
| Very early PTD, <28 weeks | 820 (0.2) | 55 (0.5) | 1.73 (1.31–2.28) | **<0.001** | 7 (0.3) | 0.97 (0.46–2.06) | 0.94 | 139 (0.6) | 2.20 (1.82–2.66) | **<0.001** | 87 (0.3) | 1.10 (0.87–1.39) | 0.42 |
| Spontaneous PTD | 11,409 (3.4) | 493 (4.2) | 1.18 (1.07–1.29) | **0.001** | 100 (3.9) | 1.17 (0.95–1.43) | 0.14 | 1,699 (7.3) | 2.06 (1.95–2.17) | **<0.001** | 1,291 (3.8) | 1.14 (1.07–1.21) | **<0.001** |
| pPROM | 5,110 (1.5) | 232 (2.0) | 1.22 (1.07–1.40) | **0.004** | 64 (2.5) | 1.52 (1.18–1.96) | **0.001** | 934 (4.0) | 2.36 (2.19–2.54) | **<0.001** | 521 (1.5) | 1.09 (0.99–1.20) | 0.09 |
| PROM in deliveries at ≥37 weeks | 21,906 (6.8) | 828 (7.5) | 1.06 (0.98–1.14) | 0.13 | 251 (10.4) | 1.24 (1.08–1.42) | **0.002** | 1,772 (8.4) | 1.11 (1.05–1.17) | **<0.001** | 1,719 (5.4) | 0.99 (0.94–1.04) | 0.64 |
| SGA[2] | 6,873 (2.0) | 320 (2.7) | 1.09 (0.97–1.22) | 0.16 | 65 (2.6) | 0.96 (0.75–1.24) | 0.76 | 594 (2.6) | 1.02 (0.94–1.12) | 0.64 | 715 (2.1) | 0.99 (0.91–1.07) | 0.72 |
| Apgar score < 7 at 5 minutes | 4,165 (1.2) | 191 (1.6) | 1.20 (1.04–1.40) | **0.014** | 41 (1.6) | 1.04 (0.76–1.43) | 0.79 | 366 (1.6) | 1.14 (1.02–1.27) | **0.019** | 323 (1.0) | 0.89 (0.79–1.00) | **0.043** |
| Neonatal mortality | 343 (0.1) | 24 (0.2) | 1.81 (1.19–2.76) | **0.006** | 7 (0.3) | 2.69 (1.25–5.78) | **0.011** | 47 (0.2) | 1.79 (1.30–2.45) | **<0.001** | 29 (0.1) | 0.71 (0.47–1.05) | 0.09 |
| Intrauterine fetal death | 711 (0.2) | 43 (0.4) | 1.55 (1.13–2.12) | **0.006** | 6 (0.2) | 0.93 (0.41–2.09) | 0.86 | 74 (0.3) | 1.35 (1.05–1.72) | **0.019** | 50 (0.1) | 0.71 (0.53–0.96) | **0.026** |
| Chorioamnionitis | 895 (0.3) | 45 (0.4) | 1.25 (0.92–1.69) | 0.15 | 10 (0.4) | 1.00 (0.53–1.88) | 1.00 | 196 (0.8) | 2.75 (2.33–3.23) | **<0.001** | 74 (0.2) | 1.02 (0.80–1.31) | 0.85 |
| Intrapartum fever | 2,189 (0.6) | 87 (0.7) | 1.01 (0.82–1.26) | 0.90 | 37 (1.5) | 1.40 (1.00–1.96) | **0.050** | 213 (0.9) | 1.24 (1.07–1.44) | **0.003** | 133 (0.4) | 0.89 (0.74–1.06) | 0.19 |
| Neonatal sepsis | 2,508 (0.7) | 97 (0.8) | 0.99 (0.80–1.21) | 0.89 | 14 (0.6) | 0.67 (0.40–1.14) | 0.14 | 300 (1.3) | 1.55 (1.37–1.75) | **<0.001** | 216 (0.6) | 0.86 (0.74–0.99) | **0.041** |

Statistically significant p-values in bold. Analyses adjusted for year of delivery, maternal age, parity, BMI, marital status, country of birth, infant's sex, smoking, income, education level, and assisted reproduction.

[1]Analyses compared to reference group 2007–2016.

[2]Missing data: reference group, n = 575; HPV infection (cytology) group, n = 24; HPV infection (HPV test) group, n = 3; treated group, n = 47; subsequent CIN2 + group, n = 71.

aOR, adjusted odds ratio; CI, confidence interval; CIN, cervical intraepithelial neoplasia; HPV, human papillomavirus; pPROM, preterm prelabor rupture of membranes; PROM, prelabor rupture of membranes; PTD, preterm delivery; SGA, small for gestational age.

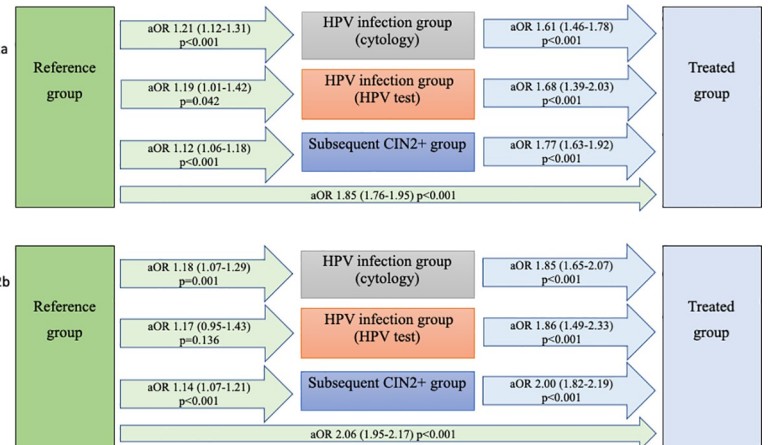

**Fig 2. Risk of preterm delivery and spontaneous preterm delivery—adjusted logistic regression.** Preterm delivery (2a); spontaneous preterm delivery (2b). Women with HPV infection had an increased risk of preterm delivery and spontaneous preterm delivery; treatment increased the risk further. aORs are given, with 95% confidence intervals in parentheses. Analyses adjusted for year of delivery, maternal age, parity, BMI, marital status, country of birth, infant's sex, smoking, income, education level, and assisted reproduction. aOR, adjusted odds ratio; CIN, cervical intraepithelial neoplasia; HPV, human papillomavirus.

## SGA and Apgar score

In the study population, 41,323 infants were born SGA (2.3%). Data were missing for 4,028 deliveries (0.2%). There was no difference between the reference group and any of the exposure groups in regard to SGA risk (Table 2).

In the study population, 22,968 babies were born with a 5-minute Apgar score < 7 (1.3%). The treated group had higher risk of low Apgar score compared to the reference group (Table 2) and to the subsequent CIN2+ group (Table 4). Furthermore, the HPV infection (cytology) group had an increased risk of low Apgar score. The association was oriented in the same direction in the HPV infection (HPV test) group but did not reach significance (Table 2).

## Intrauterine fetal death

There were 5,649 (0.3%) reported cases of intrauterine fetal death in the study population. Compared to the reference group (0.2%), the risk was increased in the HPV infection (cytology) group (0.4%, aOR 1.55, 95% CI 1.13–2.12, $p = 0.006$) and in the treated group (0.3%, aOR 1.35, 95% CI 1.05–1.72, $p = 0.019$), while it was decreased in the subsequent CIN2+ group (0.1%, aOR 0.71, 95% CI 0.53–0.96, $p = 0.026$) (Table 2). This decreased risk was significant for the subsequent CIN2+ subgroup of women who were diagnosed >3 years after delivery (Table 3).

## Infectious complications

In the study population, 5,170 (0.3%) deliveries were complicated by chorioamnionitis, 10,832 (0.6%) by intrapartum fever, and 12,923 (0.7%) by neonatal sepsis. The treated group had an increased risk of chorioamnionitis (0.8%, aOR 2.75, 95% CI 2.33–3.23, $p < 0.001$), intrapartum fever (0.9%, aOR 1.24, 95% CI 1.07–1.44, $p = 0.003$), and neonatal sepsis (1.3%, aOR 1.55, 95% CI 1.37–1.75, $p < 0.001$) compared to the reference group (Table 2).

The treated group experienced PTD, pPROM, and infectious complications more frequently than the reference group. For a depiction of the relationship between PTD, pPROM, and infectious complications in the treated and reference groups, see Fig 4.

**Table 3. Obstetric and neonatal outcomes in the subsequent CIN2+ group compared with the reference group, stratified for interval from delivery to CIN2+ diagnosis—adjusted multivariable logistic regression analyses.**

| Outcome | Reference group (n = 338,109) | Subsequent CIN2+ group | | | | | | Comparison of subsequent CIN2+ ≤3 years and >3 years after delivery |
| | | ≤3 years after delivery (n = 16,152) | | | >3 years after delivery[1] (n = 17,608) | | | |
| | n (%) | n (%) | aOR (95% CI)[2] | p-Value | n (%) | aOR (95% CI)[2] | p-Value | p-Value |
| PTD, <37 weeks | 15,661 (4.6) | 890 (5.5) | 1.18 (1.10–1.27) | **<0.001** | 846 (4.8) | 1.06 (0.99–1.15) | 0.11 | **0.038** |
| Early PTD, <34 weeks | 4,221 (1.2) | 249 (1.5) | 1.22 (1.07–1.39) | **0.003** | 239 (1.4) | 1.13 (0.99–1.30) | 0.08 | 0.44 |
| Very early PTD, <28 weeks | 820 (0.2) | 45 (0.3) | 1.13 (0.84–1.54) | 0.42 | 42 (0.2) | 1.07 (0.77–1.48) | 0.71 | 0.78 |
| Spontaneous PTD | 11,409 (3.4) | 665 (4.1) | 1.20 (1.11–1.30) | **<0.001** | 626 (3.6) | 1.07 (0.98–1.16) | 0.14 | **0.038** |
| pPROM | 5,110 (1.5) | 288 (1.8) | 1.17 (1.04–1.33) | **0.010** | 233 (1.3) | 0.99 (0.86–1.14) | 0.88 | 0.06 |
| PROM in deliveries at ≥37 weeks | 21,906 (6.8) | 927 (6.1) | 0.98 (0.92–1.05) | 0.65 | 792 (4.7) | 0.99 (0.92–1.07) | 0.82 | 0.89 |
| SGA[3] | 6,873 (2.0) | 355 (2.2) | 0.98 (0.88–1.10) | 0.78 | 360 (2.1) | 0.99 (0.88–1.11) | 0.84 | 0.96 |
| Apgar score < 7 at 5 minutes | 4,165 (1.2) | 177 (1.1) | 0.93 (0.80–1.09) | 0.39 | 146 (0.8) | 0.83 (0.70–0.98) | **0.032** | 0.29 |
| Neonatal mortality | 343 (0.1) | 17 (0,1) | 0.95 (0.58–1.55) | 0.82 | 12 (0.1) | 0.51 (0.28–0.93) | **0.027** | 0.10 |
| Intrauterine fetal death | 711 (0.2) | 34 (0.2) | 0.99 (0.70–1.41) | 0.97 | 16 (0.1) | 0.44 (0.26–0.73) | **0.001** | **0.007** |
| Chorioamnionitis | 895 (0.3) | 40 (0.2) | 1.03 (0.75–1.42) | 0.86 | 34 (0.2) | 1.02 (0.71–1.46) | 0.93 | 0.96 |
| Intrapartum fever | 2,189 (0.6) | 83 (0.5) | 0.93 (0.75–1.17) | 0.54 | 50 (0.3) | 0.81 (0.61–1.09) | 0.17 | 0.46 |
| Neonatal sepsis | 2,508 (0.7) | 124 (0.8) | 1.00 (0.83–1.20) | 0.96 | 92 (0.5) | 0.71 (0.57–0.89) | **0.003** | **0.018** |

Statistically significant p-values in bold. Analyses adjusted for year of delivery, maternal age, parity, BMI, marital status, country of birth, infant's sex, smoking, income, education level, and assisted reproduction.

[1] Up to 18 years after delivery.

[2] Comparison with reference group.

[3] Missing data: reference group, n = 575; subsequent CIN2+ ≤3 years after delivery, n = 25; subsequent CIN2+ >3 years after delivery, n = 46.

aOR, adjusted odds ratio; CI, confidence interval; CIN, cervical intraepithelial neoplasia; HPV, human papillomavirus; pPROM, preterm prelabor rupture of membranes; PROM, prelabor rupture of membranes; PTD, preterm delivery; SGA, small for gestational age.

After stratification for PTD or not, the increased risk of infectious complications in the treated versus the reference group was still statistically significant in the PTD group (chorioamnionitis: aOR 3.96, 95% CI 3.13–5.02, p < 0.001; neonatal sepsis: aOR 1.74, 95% CI 1.43–2.12, p < 0.001), but not in the term delivery group (chorioamnionitis: aOR 1.28, 95% CI 0.98–1.67, p = 0.07; neonatal sepsis: aOR 1.08, 95% CI 0.91–1.29, p = 0.36). There was no clear increase in infectious complications in the HPV infection groups (Table 2).

Stratifying for pPROM in the PTD cases revealed that the aOR for chorioamnionitis was higher in the treated group than in the reference group, in both women with pPROM (aOR 3.68, 95% CI 2.70–5.03, p < 0.001) and women without pPROM (aOR 2.19, 95% CI 1.32–3.61, p = 0.002), but was significantly higher in women with pPROM (p = 0.032). Similarly, the increase in risk of neonatal sepsis in the treated group compared to the reference group was

**Table 4. Obstetric and neonatal outcomes in the treated group compared to the HPV infection groups and the subsequent CIN2+group—adjusted multivariable logistic regression analyses.** .

| Outcome | Treated versus HPV infection (cytology) comparison | | | | Treated versus HPV infection (HPV test) comparison | | | | Treated versus subsequent CIN2 + comparison | | | |
|---|---|---|---|---|---|---|---|---|---|---|---|---|
| | Treated group (n = 22,711) | HPV infection (cytology) group (n = 11,727) | | | Treated group (n = 14,579) | HPV infection (HPV test) group (n = 2,550) | | | Treated group, (n = 18,505) | Subsequent CIN2+ group (n = 33,760) | | |
| | n (%) | n (%) | aOR (95% CI) | p-Value | n (%) | n (%) | aOR[1] (95% CI) | p-Value | n (%) | n (%) | aOR (95% CI) | p-Value |
| PTD, <37 weeks | 2,066 (9.1) | 692 (5.9) | 1.61 (1.46–1.78) | **<0.001** | 1,313 (9.0) | 143 (5.6) | 1.68 (1.39–2.03) | **<0.001** | 1,751 (9.5) | 1,736 (5.1) | 1.77 (1.63–1.92) | **<0.001** |
| Early PTD, <34 weeks | 645 (2.8) | 221 (1.9) | 1.53 (1.29–1.81) | **<0.001** | 416 (2.9) | 34 (1.3) | 2.18 (1.50–3.17) | **<0.001** | 554 (3.0) | 488 (1.4) | 1.85 (1.59–2.13) | **<0.001** |
| Very early PTD, <28 weeks | 136 (0.6) | 55 (0.5) | 1.32 (0.93–1.87) | 0.12 | 98 (0.7) | 7 (0.3) | 2.79 (1.23–6.35) | **0.014** | 118 (0.6) | 87 (0.3) | 2.01 (1.44–2.80) | **<0.001** |
| Spontaneous PTD | 1,669 (7.3) | 493 (4.2) | 1.85 (1.65–2.07) | **<0.001** | 1,038 (7.1) | 100 (3.9) | 1,86 (1.49–2.33) | **<0.001** | 1,421 (7.7) | 1,291 (3.8) | 2.00 (1.82–2.19) | **<0.001** |
| pPROM | 918 (4.0) | 232 (2.0) | 2.09 (1.78–2.45) | **<0.001** | 598 (4.1) | 64 (2.5) | 1.55 (1.17–2.04) | **0.002** | 779 (4.2) | 521 (1.5) | 2.51 (2.20–2.87) | **<0.001** |
| PROM in deliveries at ≥37 weeks | 1,752 (8.5) | 828 (7.5) | 1.08 (0.98–1.18) | 0.12 | 1,208 (9.1) | 251 (10.4) | 0.88 (0.75–1.03) | 0.11 | 1,526 (9.1) | 1,719 (5.4) | 1.21 (1.11–1.31) | **<0.001** |
| SGA[2] | 587 (2.6) | 320 (2.7) | 0.86 (0.74–1.01) | 0.06 | 383 (2.6) | 65 (2.6) | 0.93 (0.70–1.25) | 0.64 | 523 (2.8) | 715 (2.1) | 1.03 (0.90–1.17) | 0.72 |
| Apgar score <7 at 5 minutes | 355 (1.6) | 191 (1.6) | 0.94 (0.78–1.14) | 0.55 | 248 (1.7) | 41 (1.6) | 1.16 (0.81–1.67) | 0.43 | 303 (1.6) | 323 (1.0) | 1.30 (1.08–1.57) | **0.005** |
| Neonatal mortality | 43 (0.2) | 24 (0.2) | 1.04 (0.59–1.83) | 0.89 | 21 (0.1) | 7 (0.3) | 0.56 (0.21–1.45) | 0.23 | 35 (0.2) | 29 (0.1) | 2.61 (1.45–4.70) | **0.001** |
| Intrauterine fetal death | 71 (0.3) | 43 (0.4) | 0.72 (0.48–1.09) | 0.12 | 51 (0.3) | 6 (0.2) | 1.21 (0.49–3.00) | 0.69 | 58 (0.3) | 50 (0.1) | 1.75 (1.13–2.73) | **0.013** |
| Chorioamnionitis | 192 (0.8) | 45 (0.4) | 2.21 (1.55–3.15) | **<0.001** | 119 (0.8) | 10 (0.4) | 2.47 (1.24–4.91) | **0.010** | 160 (0.9) | 74 (0.2) | 3.33 (2.41–4.60) | **<0.001** |
| Intrapartum fever | 211 (0.9) | 87 (0.7) | 1.21 (0.92–1.59) | 0.17 | 154 (1.1) | 37 (1.5) | 0.79 (0.53–1.17) | 0.24 | 197 (1.1) | 133 (0.4) | 1.37 (1.06–1.77) | **0.015** |
| Neonatal sepsis | 294 (1.3) | 97 (0.8) | 1.61 (1.25–2.07) | **<0.001** | 171 (1.2) | 14 (0.5) | 2.34 (1.30–4.24) | **0.005** | 257 (1.4) | 216 (0.6) | 2.02 (1.63–2.51) | **<0.001** |

Statistically significant p-values in bold. Analyses adjusted for year of delivery, maternal age, parity, BMI, marital status, country of birth, infant's sex, smoking, income, education level, and assisted reproduction.

[1]Analyses 2007–2016.

[2]Missing data: treated group, n = 47; HPV infection (cytology) group, n = 24; HPV infection (HPV test) group, n = 3; subsequent CIN2+ group, n = 71.

aOR, adjusted odds ratio; CI, confidence interval; CIN, cervical intraepithelial neoplasia; HPV, human papillomavirus; pPROM, preterm prelabor rupture of membranes; PROM, prelabor rupture of membranes; PTD, preterm delivery; SGA, small for gestational age.

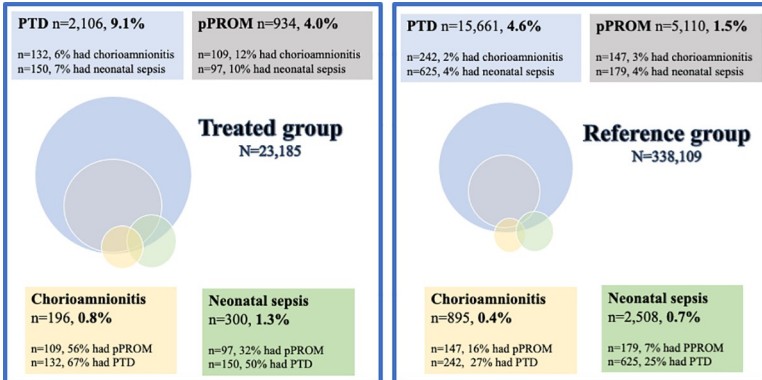

**Fig 3. Risk of pPROM—adjusted logistic regression.** Women with HPV infection had an increased risk of pPROM; treatment increased the risk further. aORs are given, with 95% confidence intervals in parentheses. Analyses adjusted for year of delivery, maternal age, parity, BMI, marital status, country of birth, infant's sex, smoking, income, education level, and assisted reproduction. aOR, adjusted odds ratio; CIN, cervical intraepithelial neoplasia; HPV, human papillomavirus; pPROM, preterm prelabor rupture of membranes.

significantly higher in women with PTD with pPROM (aOR 2.70, 95% CI 1.99–3.68, $p <$ 0.001) than in women with PTD without pPROM (aOR 0.86, 95% CI 0.62–1.20, $p = 0.39$) ($p = 0.015$ for the interaction).

## Neonatal mortality

There were 2,657 reported cases of neonatal mortality (0.1%). Neonatal mortality was increased in the treated group (aOR 1.79, 95% CI 1.30–2.45, $p < 0.001$), as well as in the HPV infection (cytology) (aOR 1.81, 95% CI 1.19–2.76, $p = 0.006$) and HPV infection (HPV test) (aOR 2.69, 95% CI 1.25–5.78, $p = 0.011$) groups, compared to the reference group (Table 2).

After additional adjustment for PTD, the increased risk of neonatal mortality disappeared in the treated group, suggesting that the risk increase had been mediated by gestational age. However, the risk was still increased in the HPV infection groups, i.e., cytology (aOR 1.64, 95% CI 1.07–2.50, $p = 0.024$) and HPV test (aOR 2.30, 95% CI 1.06–4.99, $p = 0.036$).

Of those with neonatal mortality, 4.1% in the reference group and 14.9% in the treated group also contracted chorioamnionitis, and 6.4% in the reference group and 25.5% in the treated group had infants with neonatal sepsis. For an illustration of the relationship between neonatal mortality, infectious complications, and PTD, see Fig 5. The increase in risk of

**Fig 4. Venn diagram illustrating the relationship between preterm delivery (PTD), preterm prelabor rupture of membranes (pPROM), and infectious complications in the treated group and the reference group.** The treated group had PTD, pPROM, and infectious complications more frequently. PTD and deliveries after pPROM were more often complicated by infections in the treated group than in the reference group. The deliveries with infectious complications were more frequently preterm and complicated by pPROM in the treated group than in the reference group.

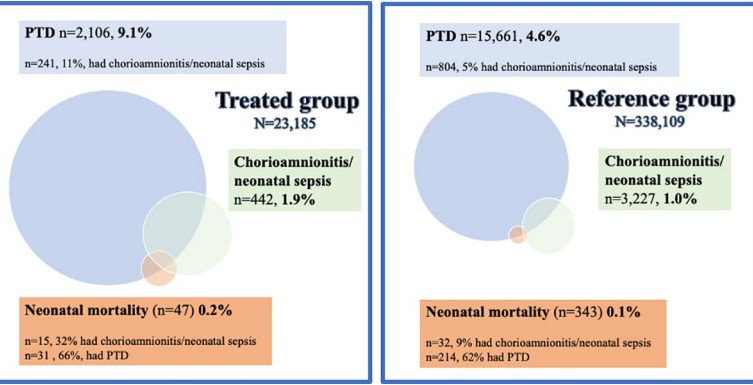

**Fig 5. Venn diagram illustrating the relationship between preterm delivery (PTD), infectious complications (chorioamnionitis/neonatal sepsis), and neonatal mortality in the treated group and the reference group.** The treated group more frequently had PTD, neonatal mortality, and infectious complications. The deliveries resulting in neonatal mortality more frequently had infectious complications in the treated group than in the reference group.

neonatal mortality in the treated group compared to the reference group was significantly higher in women with infectious complications (chorioamnionitis or neonatal sepsis) (aOR 2.06, 95% CI 1.00–4.24, $p = 0.049$) than in women without infectious complications (aOR 0.90, 95% CI 0.60–1.34, $p = 0.59$) ($p = 0.015$ for the interaction).

## Paired analyses

In the paired analyses of 4,680 women with deliveries both before and after treatment, the risk of PTD (aOR 1.26, 95% CI 1.07–1.49, $p = 0.007$), spontaneous PTD (aOR 1.23, 95% CI 1.02–1.49, $p = 0.032$), and pPROM (aOR 1.72, 95% CI 1.29–2.29, $p < 0.001$) were increased after treatment (Table 5). Also, chorioamnionitis and neonatal sepsis had higher odds ratios after treatment (Table 5).

## HPV infection groups

Except for a higher risk of pPROM (aOR 1.43 95% CI 1.03–1.99, $p = 0.034$) and PROM (aOR 1.26, 95% CI 1.06–1.50, $p = 0.008$) in the HPV infection (HPV test) group compared to the HPV infection (cytology) subgroup with no HPV test, no significant differences in outcomes were found between the HPV infection groups (S8 Table).

## Discussion

### Main findings

In this population-based study, women with HPV infection shortly before or during pregnancy had an increased risk of PTD, spontaneous PTD, pPROM, PROM, and neonatal mortality. An even higher risk of PTD, spontaneous PTD, and pPROM was found in women with previous treatment for CIN. Previous treatment was also associated with PROM, neonatal mortality, and maternal and neonatal infectious complications.

### Comparison with previous studies and interpretation

**Treatment for CIN and obstetric and neonatal outcomes.** The finding that treatment for CIN is associated with infectious pregnancy complications has not been clearly demonstrated before, but an increased risk of chorioamnionitis was suggested by a recent meta-

**Table 5. Obstetric and neonatal outcomes in paired analyses of a woman's last delivery before treatment and first delivery after treatment—conditional logistic regression analysis (*n* = 4,680 women).**

| Outcome | Before treatment | After treatment | Unadjusted analyses | | Adjusted analyses[1] | |
|---|---|---|---|---|---|---|
| | *n* (%) | *n* (%) | OR (95% CI) | *p*-Value | aOR (95% CI) | *p*-Value |
| PTD, <37 weeks | 288 (6.2) | 355 (7.6) | 1.23 (1.06–1.44) | **0.008** | 1.26 (1.07–1.49) | **0.007** |
| Early PTD, <34 weeks | 72 (1.5) | 107 (2.3) | 1.49 (1.10–2.00) | **0.009** | 1.61 (1–12–2.33) | **0.010** |
| Very early PTD, <28 weeks | 11 (0.2) | 21 (0.4) | 1.91 (0.92–3.96) | 0.08 | 4.50 (0.90–22.45) | 0.07 |
| Spontaneous PTD | 227 (4.9) | 278 (5.9) | 1.23 (1.03–1.46) | **0.023** | 1.23 (1.02–1.49) | **0.032** |
| pPROM | 86 (1.8) | 155 (3.3) | 1.80 (1.39–2.35) | **<0.001** | 1.72 (1.29–2.29) | **<0.001** |
| PROM in deliveries at ≥ 37 weeks | 241 (5.5) | 246 (5.7) | 1.02 (0.85–1.22) | 0.85 | 1.04 (0.85–1.26) | 0.74 |
| SGA[2] | 113 (2.4) | 71 (1.5) | 0.63 (0.47–0.85) | **0.002** | 0.70 (0.50–0.99) | **0.043** |
| Apgar score < 7 at 5 minutes | 44 (0.9) | 63 (1.3) | 1.43 (0.97–2.10) | 0.068 | 1.29 (0.82–2.03) | 0.28 |
| Neonatal mortality | 6 (0.1) | 12 (0.3) | 2.00 (0.75–5.33) | 0.17 | —[3] | —[3] |
| Intrauterine fetal death | 9 (0.2) | 16 (0.3) | 1.78 (0.79–4.02) | 0.17 | —[3] | —[3] |
| Chorioamnionitis | 5 (0.1) | 36 (0.8) | 7.20 (2.83–18.35) | **<0.001** | 14.22 (2.09–96.92) | **0.007** |
| Intrapartum fever | 24 (0.5) | 16 (0.3) | 0.67 (0.35–1.26) | 0.21 | 0.83 (0.35–1.94) | 0.66 |
| Neonatal sepsis | 31 (0.7) | 43 (0.9) | 1.39 (0.87–2.20) | 0.17 | 1.92 (1.02–3.61) | **0.044** |

Statistically significant *p*-values in bold.

[1]Adjusted for BMI, marital status, infant's sex, smoking, and assisted reproduction.

[2]Missing data about SGA for 7 women before treatment and 4 women after treatment.

[3]Too few cases to adjust for in multivariable model.

aOR, adjusted odds ratio; CI, confidence interval; HPV, human papillomavirus; OR, odds ratio; pPROM, preterm prelabor rupture of membranes; PROM, prelabor rupture of membranes; PTD, preterm delivery; SGA, small for gestational age.

analysis [10]. To the best of our knowledge, this is the first study to demonstrate that it is not only the risk of pPROM but also the risk of PROM at term that is increased after treatment for CIN. Infection is considered to be an important cause of PTD and pPROM [9], while deliveries after pPROM and PROM also entail increased risk of infectious complications [33]. Our results could be explained by an increased risk of ascending bacterial infection in women previously treated for CIN, causing PTD, pPROM, chorioamnionitis, and neonatal sepsis. The finding of an increased risk of PTD, including pPROM, after treatment is consistent with previous studies [10]. The risk increase for PTD was similar to results from birth cohort studies in Finland (1998–2009) and Denmark (1997–2005) [12,14]. The first systematic review that highlighted the risk of PTD after treatment for CIN was published in 2006 [34]. In our study, 40% of women treated for CIN were treated after 2006. Eighty-nine percent were treated after 1996, indicating that the majority had been treated with the LLETZ technique. The results of this study thus reflect the risks related to current treatment methods.

**HPV infection and obstetric and neonatal outcomes.** In this study, women with HPV infection had increased risk of PTD, pPROM, PROM, and neonatal mortality. Previous studies have presented conflicting results. A recent population-based study did not confirm an increased risk of PTD in women positive for high-risk HPV without CIN2+ (aOR 1.11, 95% CI 0.75–1.66) [20], perhaps due to a lack of power based on the much smaller sample. The risk estimates in our study (aOR 1.21 for cytology and aOR 1.19 for HPV test) were slightly lower than that suggested by a recent meta-analysis (aOR 1.50), but within the 95% CI of that study (1.19–1.88) [22]. HPV has been found in the placenta [19,35], and it has been hypothesized that HPV might cause placental dysfunction, leading to PTD [18,19]. However, in this study we found no increase in SGA in the exposed women, suggesting a mechanism other than dysfunctional placenta. In a mouse model, viral infection of the cervix during pregnancy reduced

the capacity of the female reproductive tract to prevent ascending bacterial infection [36]. We did not confirm any association between HPV infection and clinical infectious complications. However, infections associated with PTD are known to be subclinical to a major extent [37].

It is still unclear whether the HPV infection itself, common confounders, or a genetic predisposition is the causal link behind the association between CIN and PTD. However, our results suggest that HPV infection might be an important factor. Women developing CIN2 + after delivery also had a small increased risk of PTD and spontaneous PTD, although it was not significant in the subgroup of women who were diagnosed more than 3 years after delivery. This may possibly reflect an ongoing HPV infection during pregnancy and delivery in women with earlier diagnosis of CIN2+ after birth. These findings contradict the hypothesis of a common genetic susceptibility to both persistent HPV infection and PTD or an association merely due to confounding.

## Strengths and limitations

To the best of our knowledge, this is the largest study to date examining the associations of HPV infection shortly before or during pregnancy and treatment for CIN with obstetric and neonatal outcomes. It comprises an entire population, and we believe it compares the largest number of deliveries before and after treatment for CIN in the same women published so far.

As this is an observational study, causality cannot be established and residual confounding cannot be ruled out despite thorough adjustments. However, the risk increases for PTD, spontaneous PTD, pPROM, and chorioamnionitis found in the treated group were supported by similar findings in the paired analyses, in which 2 deliveries in the same woman were compared.

The definition of HPV infection in this study has some potential limitations. To begin with, relying on positive screening tests may lead to an underestimation of the true incidence and prevalence as many infections can occur and resolve between testing. A small minority of HPV tests might have been positive due to exclusive detection of low-risk strains, although the tests used at Swedish laboratories predominantly specifically detect high-risk HPV. Since HPV test results only became available in 2007, abnormal cytology was used as a proxy for HPV infection, as in earlier studies [21,38]. A Swedish study found that 72% of women aged under 40 years with low-grade abnormal cytology were positive for high-risk HPV [39]. Since the abnormal cytology group in this study also included high-grade abnormality, abnormal cytology can be assumed to represent the presence of high-risk HPV to a high degree. Positive HPV tests/cytology up to 6 months before conception were included in the HPV infection groups, and some infections might have resolved before pregnancy. However, such misclassification would have attenuated a true effect. We did not require a negative HPV test in the reference group, and some women might have had an undetected HPV infection. However, such misclassification would have attenuated the associations found for the exposure groups compared to the reference group. Since no national treatment data are available, CIN3 was used as a proxy for treatment; all women treated for high-grade lesions are thus not included in the treated group. For the paired analyses it was not possible to adjust for maternal age and parity, and these results must thus be interpreted with caution.

## Implications and next step for research

The use of HPV tests in cervical cancer screening has increased during the last decade, which will facilitate large population-based analysis of the relationship of HPV infection and pregnancy outcomes in the near future. Our results support general HPV vaccination programs, and future studies in vaccinated cohorts will ascertain whether PTD and neonatal mortality

decrease. Additional experimental studies are also needed to establish the causal pathways linking HPV infection and treatment for CIN to PTD and other adverse obstetric outcomes. This may be a key to understanding the biological mechanism leading to PTD. We suggest that pregnancies after treatment for CIN should be regarded as high-risk pregnancies.

## Conclusion

This large population-based study demonstrates that women with HPV infection shortly before or during pregnancy have an increased risk of PTD, pPROM, PROM, and neonatal mortality. Treatment for CIN was associated with further increased risk of PTD and pPROM and was also associated with PROM, neonatal mortality, and an increased risk of maternal and neonatal infectious complications. These results suggest that pregnancies after treatment for CIN should be regarded as high-risk pregnancies and counseled accordingly. Furthermore, our results support general HPV vaccination programs.

## Supporting information

**S1 STROBE Checklist.**
(DOC)

**S1 Table. Diagnosis codes, according to the International Classification of Diseases–10th Revision (ICD-10), registered in the Swedish Medical Birth Register, leading to exclusion.**
(DOC)

**S2 Table. Definition of study groups.**
(DOC)

**S3 Table. Classification of cervical cytology and histology.**
(DOC)

**S4 Table. Outcome definition based on diagnosis codes according to the International Classification of Diseases–10th Revision (ICD-10) in the Swedish Medical Birth Register.**
(DOC)

**S5 Table. Obstetric and neonatal outcomes in exposure groups compared to the reference group—univariable logistic regression analyses.**
(DOC)

**S6 Table. Obstetric and neonatal outcomes in the subsequent CIN2+ group compared with the reference group, stratified by interval from delivery to CIN2+ diagnosis—univariable logistic regression analyses.**
(DOC)

**S7 Table. Obstetric and neonatal outcomes in the treated group compared to the HPV infection groups and the subsequent CIN2+group—univariable logistic regression analyses.**
(DOC)

**S8 Table. Outcomes in the HPV infection (cytology) group compared to the HPV infection (HPV test) group—univariable and multivariable logistic regression analyses, deliveries 2007–2016.**
(DOC)

## Author Contributions

**Conceptualization:** Johanna Wiik, Cecilia Kärrberg, Björn Strander, Bo Jacobsson, Verena Sengpiel.

**Data curation:** Johanna Wiik, Staffan Nilsson, Björn Strander, Verena Sengpiel.

**Formal analysis:** Johanna Wiik, Staffan Nilsson.

**Funding acquisition:** Johanna Wiik, Verena Sengpiel.

**Methodology:** Staffan Nilsson, Cecilia Kärrberg, Björn Strander, Bo Jacobsson, Verena Sengpiel.

**Project administration:** Johanna Wiik, Verena Sengpiel.

**Supervision:** Staffan Nilsson, Verena Sengpiel.

**Writing – original draft:** Johanna Wiik, Verena Sengpiel.

**Writing – review & editing:** Johanna Wiik, Staffan Nilsson, Cecilia Kärrberg, Björn Strander, Bo Jacobsson, Verena Sengpiel.

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
