## [Editor Report · Decision Letter 0]

7 Jan 2021

Dear Dr Wiik, 

Thank you for submitting your manuscript entitled "Treated and untreated human papillomavirus infection is associated with preterm delivery and neonatal mortality - a Swedish population-based study" for consideration by PLOS Medicine.

Your manuscript has now been evaluated by the PLOS Medicine editorial staff and I am writing to let you know that we would like to send your submission out for external peer review.

Kind regards,

Caitlin Moyer, Ph.D.

Associate Editor

PLOS Medicine

---

## [Decision Letter · Decision Letter 1]

16 Feb 2021

Dear Dr. Wiik,

Thank you very much for submitting your manuscript "Treated and untreated human papillomavirus infection is associated with preterm delivery and neonatal mortality - a Swedish population-based study" (PMEDICINE-D-20-06245R1) for consideration at PLOS Medicine. 

Your paper was evaluated by a senior editor and discussed among all the editors here. It was also discussed with an academic editor with relevant expertise, and sent to three independent reviewers, including a statistical reviewer. The reviews are appended at the bottom of this email and any accompanying reviewer attachments can be seen via the link below:

[LINK]

In light of these reviews, I am afraid that we will not be able to accept the manuscript for publication in the journal in its current form, but we would like to consider a revised version that addresses the reviewers' and editors' comments. Obviously we cannot make any decision about publication until we have seen the revised manuscript and your response, and we plan to seek re-review by one or more of the reviewers. 

We expect to receive your revised manuscript by Mar 09 2021 11:59PM. Please email us (plosmedicine@plos.org) if you have any questions or concerns.

We look forward to receiving your revised manuscript. 

Sincerely,

Caitlin Moyer, Ph.D.

Associate Editor 

PLOS Medicine

plosmedicine.org

1.Title: Please revise your title according to PLOS Medicine's style. Your title must be nondeclarative and not a question. It should begin with main concept if possible. "Effect of" should be used only if causality can be inferred, i.e., for an RCT. Please place the study design ("A randomized controlled trial," "A retrospective study," "A modelling study," etc.) in the subtitle (ie, after a colon). We suggest: “Associations of reated and untreated human papillomavirus infection with preterm delivery and neonatal mortality: A Swedish population-based study” or similar.

2. Abstract: Methods and Findings: Please provide some of the key summary demographics of the population.

3. Abstract: Methods and Findings: Please quantify the main results (with both 95% CIs and p values).

4. Abstract: Methods and Findings: In the last sentence of the Abstract Methods and Findings section, please describe the main limitation(s) of the study's methodology.

5. Abstract: Conclusions: Please address the study implications without overreaching what can be concluded from the data; the phrase "In this study, we observed ..." may be useful.

6 .Abstract: Conclusions: Please avoid language implying causality (“Treatment for CIN increases risks ruther and seems to confer a risk…”)

7. Author summary: At this stage, we ask that you include a short, non-technical Author Summary of your research to make findings accessible to a wide audience that includes both scientists and non-scientists. The Author Summary should immediately follow the Abstract in your revised manuscript. This text is subject to editorial change and should be distinct from the scientific abstract. Please see our author guidelines for more information: https://journals.plos.org/plosmedicine/s/revising-your-manuscript#loc-author-summary

8. Main text: Throughout: Please do not include spaces within brackets of in-text citation (e.g. [2,3]) and please place the brackets before the punctuation (e.g. Page 4, line 2).

9. Introduction: Please do conclude the Introduction by mentioning your study objective/hypothesis, but this does not need to be in a separate “Objectives” section within the introduction.

10. Methods/ figure 1: Is it possible to give additional clarification/explanation of the 600,000+ excluded women (did not fulfil criteria of any group) from Fig 1 flow chart (reasons for exclusion)?

11. Methods: Please add the following statement, or similar, to the Methods: "This study is reported as per the Strengthening the Reporting of Observational Studies in Epidemiology (STROBE) guideline (S1 Checklist)."

12. Methods: Did your study have a prospective protocol or analysis plan? Please state this (either way) early in the Methods section.

13. Results: Please provide more quantitative description of the background differences in study groups (age, parity, smoking).

14. Results: For all results presented in the text, please provide both the 95% CIs and p values.

15. Results: “Small for gestational age and Apgar score” section: please clarify which groups are meant by “The groups did not differ…” in the second sentence of this paragraph.

16. Results: “HPV infection groups” section: please report the result for this association “Except for a higher risk of pPROM and PROM in the HPV infection (HPV test) group” and specify the comparison group.

17. Discussion: Please present and organize the Discussion as follows: a short, clear summary of the article's findings; what the study adds to existing research and where and why the results may differ from previous research; strengths and limitations of the study; implications and next steps for research, clinical practice, and/or public policy; one-paragraph conclusion. 

18. References: Please double check the formatting (such as use of italics- e.g. BMJ format in ref 8 and 15). Please provide an English translation for ref 26, 28, 29, 30.

19. Table 2, Table 3, Table, 4, and Table 5: Please present actual p values rather than representing with asterisks. Please provide the unadjusted results in addition to the adjusted analyses (these may be in a supporting information table, if desired).

20. Figures 2 and 3: Please include the p values with the arrows, if possible. As mentioned for the Tables, the unadjusted analyses should be included, whether in the main text or supporting information.

21. S5 Table: Please provide p values for each comparison, rather than indicating p values with asterisks. Please provide the unadjusted analyses.

22. STROBE checklist: Thank you for including the checklist. It would be appropriate to refer to locations in the text for each item with an additional column on the right-hand side of the table- When completing the checklist, please use section and paragraph numbers, rather than page numbers.

Comments from the reviewers:

Reviewer #1: This retrospective population-based register study of women with singleton deliveries registered in the Swedish Medical Birth Register 1999-2016, aims to investigate whether HPV infection in conjunction with pregnancy, as well as previous treatment for CIN, is associated with an increased risk of PTD and other adverse obstetric and neonatal outcomes.

Comments:

There are some typos in the text that need rectifying.

The STROBE checklist is suitably included in the supplementary material.

Descriptive data are appropriately presented, and the comparison of exposure groups by logistic regression (adjusted for socioeconomic and health-related confounders) is a technically valid and robust approach.

Are there other factors that may be associated with PTD that can be included as potential confounders within the models? Alcohol or drug abuse, for instance? 

Table 1 is very informative. Did the authors complete any statistical testing to explore if differences between groups are significant or not?

Table 2 is an efficient and effective way to communicate the main results.

Perhaps more caution should be applied when drawing inferences from the paired analysis. These results might not be generalisable due to the specific nature of the cohort analysed. 

For instance: Might multiple births be affecting the outcomes? Or, the effect of having treatment after at least one prior delivery? Are there other potential intra-woman confounders in this analysis (e.g age at delivery older after treatment, although it is noted that maternal age is included as a covariate in the model)?

The main limitations have been acknowledged, including the fact that this in an observational study (and therefore, causality cannot be inferred), the potential for confounding, and the possibility of misclassification.

Overall, this is a thorough study looking at different subgroups and their outcomes over a large cohort and incorporating a good use of visualisation in the figures which helps to clearly communicate the research findings.

Reviewer #2: This is a large population-based study that investigates in detail the association between HPV infection, CIN or CIN treatment and PTB. The authors have put a lot of effort and have conducted a comprehensive analysis by including all possible comparison groups (women with positive HPV testing prior to delivery; women with abnormal cytology prior to delivery; women treated for CIN3 prior to delivery; women with CIN2+ after delivery; women with no history of HPV/CIN prior to delivery) and performing all possible comparison combinations including self-matching. The authors very nicely associate their findings with previous studies, and suggest potential mechanism of PTB based on their findings. Although the main findings of this study have been suggested by other studies, findings were conflicting or not conclusive before as the authors describe in the manuscript. This is the largest and most comprehensive study so far trying to cast light on the long-standing debate whether HPV infection per se increases the risk of PTB, and therefore I highly recommend it for publication in PLoS Medicine.

I have only some minor comments to the authors:

-Line 34: Replace "outcome" with "outcomes"

-In the HPV infection group, I do not understand what the authors mean by "between six months prior to conception and delivery". I think that they mean "within six months prior to conception or during pregnancy".I suggest rephrasing throughout the text.

-Additionally, in abstract the various comparison groups are not very clear. I understand that there is a word limitation, but if possible, authors could describe the different groups in more details, especially the timing of the abnormal test/intervention. By reading the abstract only, it is not very clear for example whether the HPV positivity group includes women with HPV prior to pregnancy, during pregnancy, or both.

-Page 7, lines 16-17: I suggest adding citations for the statistical software that the authors have used.

-Page 7, line 32: Replace "Subsequent" with "subsequent"

Reviewer #3: This is a well-written substantive retrospective paper based on over 1 million births in Sweden. 

1. p.5 Line 29. Women with exclusively normal cytology. Does that mean that they had documented negative HPV tests? I ask because if they did not, how do you know that they were not also HPV positive which would put them in the comparison group. When the term Lifetime is used -what does that mean? Is it the time prior to delivery? 

2. p.6, line 13: It is possible that women who have CIN2+ at a young age can clear spontaneously. Did you exclude because they got categorized into the other exposure groups?

3. How did you account for parity? Were women who had multiple births counted as one person or was each birth counted? This could mean a women with multiple preterm births could be counted multiple times with the same characteristics over and over again.

4. P2, line 31: With the high prevalence of HPV infection in anyone who has ever been sexually active, it may be more appropriate to say, "Women with detectable HPV infection..."

[LINK]

---

## [Editor Report · Decision Letter 2]

19 Apr 2021

Dear Dr. Wiik,

Thank you very much for re-submitting your manuscript "Associations of treated and untreated human papillomavirus infection with preterm delivery and neonatal mortality: A Swedish population-based study" (PMEDICINE-D-20-06245R2) for review by PLOS Medicine.

I have discussed the paper with my colleagues and the academic editor. I am pleased to say that provided the remaining editorial and production issues are dealt with we are planning to accept the paper for publication in the journal.

The remaining issues that need to be addressed are listed at the end of this email. Please take these into account before resubmitting your manuscript:

[LINK]

In revising the manuscript for further consideration here, please ensure you address the specific points made by the editors. In your rebuttal letter you should indicate your response to the editors' comments and the changes you have made in the manuscript. Please submit a clean version of the paper as the main article file. A version with changes marked must also be uploaded as a marked up manuscript file.

We look forward to receiving the revised manuscript by Apr 26 2021 11:59PM.   

Sincerely,

Caitlin Moyer, Ph.D.

Associate Editor 

PLOS Medicine

plosmedicine.org

Requests from Editors:

1. Abstract: Methods and Findings: Please clarify if the OR reported in the text here are the adjusted ORs.

2. Abstract: Methods and Findings: Page 3, Line 1-2: We suggest revising the limitations sentence as “Limitations of the study include the retrospective design and the fact that because HPV test results only became available in 2007, abnormal cytology was used as a proxy for HPV infection.”

3. Author summary: What did the researchers do and find? In the first bullet point of this section, we suggest removing “...is, so far, the largest published and the first to demonstrate…”

4. Author summary: What do these findings mean? We suggest providing some clarification on the final bullet point in this section, for example “These results support the idea that strategies to mitigate HPV infection, such as vaccination programs, may be beneficial for maternal and neonatal pregnancy outcomes.” or similar.

5. Methods: Page 6: Lines 6-8: Please clarify this sentence to: “We did not publish the analysis plan but the overall exposure, outcomes, confounders and analyses were planned in 2015 by the research team based on hypotheses drawn from previous studies.”

6. Methods: Page 6, line 19: Please change to “30.2 years”

7. Results: Page 17, line 4: We suggest replacing “non-significant” with “not statistically significant” or similar.

8. Discussion: Page 21: Lines 25-26: We suggest revising to: “However, our results suggest that HPV infection might be an important factor.”

9. Discussion: Page 22: Line 1-2: Please revise to: “To the best of our knowledge, this is the largest study to date examining the associations of HPV infection in conjunction with pregnancy and treatment for CIN with obstetric and neonatal outcomes.” Please also temper the second sentence (compares the largest number…) at lines 3-4 with “to the best of our knowledge” or similar.

10. Discussion: Page 22, Line 11: We suggest changing “...implies an obvious underestimation…” to “...may lead to an underestimation of the true incidence…” or similar.

11. Discussion: Page 22, Line 27-28: Please revise to “For the paired analyses it was not possible to adjust for maternal age and parity and these results must thus be interpreted with caution.”

12. Discussion: conclusions: Page 23: Line 9-11: Please revise to avoid causal implications: “Treatment for CIN was associated with further increased risk of PTD, pPROM, PROM and neonatal mortality and was associated with increased risk of maternal and neonatal infectious complications. These results suggest it may be beneficial to regard pregnancy after treatment for CIN as high-risk pregnancies and counselled accordingly.”

13. References: Please check journal name abbreviations (Ref 2 should be J Natl Cancer Inst, Ref 3 should be Eur J Cancer). Please remove the italics from ref 17, and 27. Please use the "Vancouver" style for reference formatting, and see our website for other reference guidelines https://journals.plos.org/plosmedicine/s/submission-guidelines#loc-references

14. Figure 2. Please give an overall title for Figure 2, noting the differences between panels a and b. 

15. Figure 4 and Figure 5. Please provide the numbers in addition to percentages.

16. Table 5: Please note that the line numbers are overlapping with the right-side column for the last 5 rows.

17. Supporting information: If possible, please include each item as a separate file.

18. S8 Table: It is not clear what part of the table this reference is supporting (1. M. HWW. In: Wagner M MA, Aryel R. , editor. Handbook of Biosurveillance Elsevier 2006. p. 439-52.)

[LINK]

---

## [Editor Report · Decision Letter 3]

29 Apr 2021

Dear Dr Wiik, 

On behalf of my colleagues and the Academic Editor, Jenny E Myers, I am pleased to inform you that we have agreed to publish your manuscript "Associations of treated and untreated human papillomavirus infection with preterm delivery and neonatal mortality: A Swedish population-based study" (PMEDICINE-D-20-06245R3) in PLOS Medicine.

PRESS

Sincerely, 

Caitlin Moyer, Ph.D. 

Associate Editor 

PLOS Medicine